# Proactive Robot Assistance via Spatio-Temporal Object Modeling

**Maithili Patel**
Georgia Institute of Technology United States
maithili@gatech.edu

**Sonia Chernova**
Georgia Institute of Technology United States
chernova@gatech.edu

**Abstract:** Proactive robot assistance enables a robot to anticipate and provide for a user's needs without being explicitly asked. We formulate proactive assistance as the problem of the robot anticipating temporal patterns of object movements associated with everyday user routines, and proactively assisting the user by placing objects to adapt the environment to their needs. We introduce a generative graph neural network to learn a unified spatio-temporal predictive model of object dynamics from temporal sequences of object arrangements. We additionally contribute the Household Object Movements from Everyday Routines (HOMER) dataset, which tracks household objects associated with human activities of daily living across 50+ days for five simulated households. Our model outperforms the leading baseline in predicting object movement, correctly predicting locations for 11.1% more objects and wrongly predicting locations for 11.5% fewer objects used by the human user.

**Keywords:** Proactive Assistance, Robot Learning, Graph Neural Network, Spatio-Temporal Object Tracking

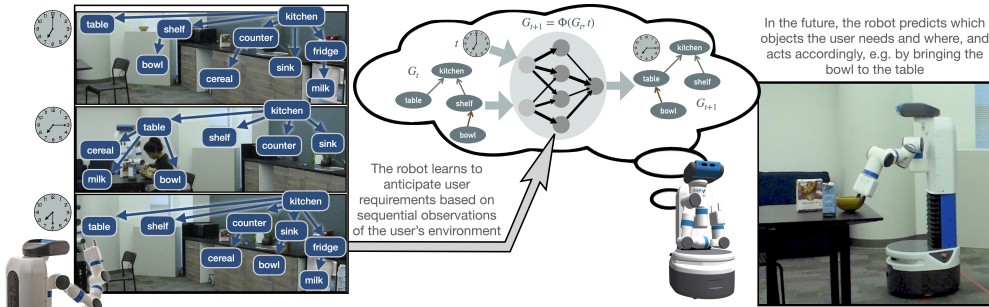

Figure 1: The robot observes and learns patterns of object movements resulting from everyday activities, which it then uses to predict and execute proactive object relocations, such that the objects are in locations where the user will need them in the near future.

## 1 Introduction

Service robots that operate in human environments and assist users with everyday tasks promise to be valuable in domestic [1] as well as workplace [2] settings, and can cater to the needs of a wide range of users, including children [3], older adults [4], and people with disabilities [5]. Critically, prior studies have shown that in longitudinal assistive scenarios, users show a strong preference for *proactive robot assistance*, such that the robot is able to anticipate and provide for the user's needs without being explicitly asked [6, 7, 8]. Prior work has focused on two proactive assistance scenarios: assistance with fixed pre-determined tasks (e.g., manufacturing assembly [9, 10], object arrangement [11]) and assistance with daily human activities in a home or office setting [7, 12, 13]

In this work, we focus on the problem of proactive robot assistance towards daily activities in the form of object placement. Examples include the robot fetching objects for the user in anticipation of their need (e.g., taking out the breakfast cereal and bowl in the morning), as well as restoring objects after use (e.g., cleaning up after a meal), all without being asked. Prior work has addressed proactive assistance by relying on detailed models of human activity, requiring the robot to perform

6th Conference on Robot Learning (CoRL 2022), Auckland, New Zealand.

human activity recognition or human action prediction in order to determine which action to perform next [12, 14, 15]. However, it may be impractical for the robot to keep the person in view continuously throughout the day, and building activity recognition models that comprehensively cover *all* human activities remains a challenging problem due to the diversity of human behavior. Moreover, since prior models are limited to assisting only with the user's current activity and predicting only a few minutes into the future, they may not provide sufficient time for a mobile manipulator to locate and fetch objects across a house.

Our work contributes a novel approach for longitudinal proactive robot assistance over large spaces (e.g. a whole house), and long time horizons (minutes to hours) without requiring recognition of the underlying user activity. Our key insight is that *objects* provide perspective into the user's activities and needs, without explicit activity recognition. Since proactive assistance primarily takes the form of object relocation, tracking the movement of objects in the environment has the benefit of providing actionable object-level information that can be utilized to provide anticipatory assistance. Therefore, as illustrated in Figure 1, our work aims to understand patterns in temporal daily routines of users, anticipate user needs, and plan assistive actions.

Specifically, the contributions of our work are as follows. First, we contribute the Household Object Movements from Everyday Routines (HOMER) dataset, which tracks household objects associated with human activities across 50+ days for five simulated households. Second, we provide a formal definition for longitudinal proactive assistance as an object dynamics modeling problem. Third, we contribute a novel generative graph neural network model that performs spatio-temporal object tracking and facilitates proactive robot assistance by modeling future movement of objects in an environment. We validate our approach against prior works that perform object tracking based on object-object relation frequencies [16] and previously observed periodic routines [17]. Our approach outperforms both baselines in predicting object movement even with as little as 5 days of training data, correctly predicting locations for 11.1% more objects and wrongly predicting locations for 11.5% fewer objects used by the human user. Additionally, we demonstrate the use of our system on a physical robot proactively assisting a user with their morning routine.

## 2 Prior Work

In this section, we discuss prior work relating to the problem of object tracking, the representation of object arrangements, and the computational models relating to our tracking formulation.

*Object tracking* methods have been developed towards manipulation in home robots [18], collision avoidance [19], augmented reality [20], robot localization [21], etc. Most closely related to our work are methods aimed at aiding object search. Probabilistic models have been used to represent beliefs over object locations by combining prior knowledge, observations, and known constraints [22], and additionally leveraging correlational information from datasets to generate semantic priors on the likelihood of inter-object relations [16]. FreMEn [17] and temporal persistence modeling [23] leverage past experience in the environment to model a temporal function of existence of objects of interest in a location. We seek to combine spatial inter-object relations and observed object location histories into a unified spatio-temporal model of object dynamics.

Object tracking for proactive assistance requires us to construct object models that span both space and time. To model space, we utilize *scene graphs*, which are represented by a set of $< object, relationship, subject >$ triplets and model their progression in time through a sequence of scene graphs at discrete time intervals. Probabilistic models [24] or graphical LSTMs [25] are typically used for inference over spatio-temporal graphs, but learning a predictive model of graphical object arrangements requires a generative model that can represent the future scene graph given the current scene graph. *Generative Graph Neural Network* models predict graphs using encoder-decoder frameworks [26, 27, 28], or step-wise modification [29, 30, 31]. We find the latter more suitable for our application, because of sparse changes in scenes over consecutive time steps. Due to lack of domain specific heuristics that some modification-based networks [29, 31] require, we derive inspiration from a general graph translation method, Node-Edge Co-evolving Deep Graph Translator (NEC-DGT) [30] to learn our predictive model only based on data. Graph Neural Networks have been adapted to model temporal dynamics [32], by exploiting relationships between graphs proximal in time [33], or related by known temporal periodicities in data [34]. However, our model also needs to derive information from the absolute value of time (e.g. the user usually has breakfast *around 8am*, in addition to *after brushing their teeth*). Hence, we use a time representation inspired from prior work [35] as a global context for our network.

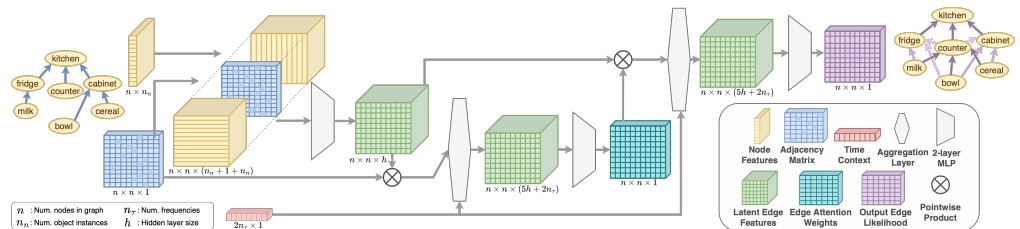

Figure 2: Generative graph neural network to predict a future scene, given the current scene and time. The input graph edges are embedded into latent space and then passed through attention-based aggregation to obtain information from neighboring edges and temporal context to predict the output edge existence.

## 3 Problem Formulation

We model the state of the environment, $X_t$, at time $t$, as a set of object-location pairs $(o_i, l_i)$ representing the placement of object $o_i \in \mathcal{O}$ at location $l_i \in \mathcal{L}$. An embodied agent (human or robot) can modify the environment by performing a *relocation* action $r(o, l_1, l_2)$ to move object $o \in \mathcal{O}$ from location $l_1$ to location $l_2$. We allow objects to serve as locations for placement of other objects (e.g., food items placed on a *plate*), such that $\mathcal{O} \cap \mathcal{L} \neq \emptyset$. We assume all objects in $\mathcal{O}$ to be relocatable, and instances of the same class (e.g., one of multiple *cereal* instances) to be uniquely identifiable.

Based on the above formulation, we model the object relocation problem as consisting of two parts. First, given a set of previous observations of the environment $X_{0:M}$ over some time span $M$, learn the model $\Phi(X_t, t) \rightarrow \hat{X}_{t+\delta}$ that takes the time $t$ and the state of the environment $X_t$ and predicts the future state $\hat{X}_{t+\delta}$ some fixed $\delta$ timesteps in the future. Second, define the function $\Psi(X_t, \hat{X}_{t+\delta}) \rightarrow \mathcal{R}$ which returns the set of relocations $\mathcal{R}$ required to transition the environment from $X_t$ to $\hat{X}_{t+\delta}$.

## 4 Spatio-Temporal Object Dynamics Model

We aim to learn the dynamics of object locations over graphical representations to conserve spatial relationship information. We represent the environment state, $X_t$ as a directed graph, $G_t = \{V_t, E_t\}$, and learn a model $\Phi(G_t, t) \rightarrow p(G_{t+\delta})$ to predict a fully-connected probabilistic graph from data. $G_t$ is an in-tree with nodes $V = \{v_i\}$ representing each object/location instances, and one edge $\{e_{i,j}\} \in E$ originating from every node, except the root node, leading to its parent location node. The output probabilistic graph $p(G_{t+\delta})$ is a distribution over in-trees, and we can infer the posterior, $\hat{G}_{t+\delta} = \arg\max_{G_{t+\delta}} \; p(G_{t+\delta})$, by picking the most likely out-edge for every node.

To learn the dynamics $\Phi$ from graph sequences, we create a graph translation model[1]. The inputs to our model are one-hot encodings identifying each object instance as node embeddings, the adjacency matrix of the input graph, and an encoding of the output graph timestamp. Our time encoding is similar to Time2Vec [35], which uses sinusoidal functions parametrized by time periods $\tau$ to generate a representation of time that can model periodicity. Instead of learning the time periods end-to-end with the prediction task, as done in Time2Vec, we use $n_\tau$ pre-specified periods to induce meaningful priors on temporal periodicities, and use both sines and cosines for our periodic functions.

Our model generates latent edge features from the input edge existence and the associated node embeddings using a Multi Layer Perceptron (MLP), and passes them through aggregation layers to predict output edge existence probabilities as outlined in Figure 2. The first aggregation step operates on the input graph topology. For every edge $e_{i,j}$, we aggregate the neighboring edge features belonging to the four categories of sharing the origin $e_{i,k}$, sharing the destination $e_{l,j}$, originating from destination $e_{j,m}$, and ending at origin $e_{n,i}$ by summation and concatenate the four resulting vectors along with the feature of edge $e_{i,j}$ and time context to produce the output feature for edge $e_{i,j}$. In this manner, the first aggregation layer generates edge features containing information about itself, its neighbors, and time, which are passed through an MLP to generate attention weights for each edge. In the second round, aggregation is done in a similar fashion, but on a fully connected graph topology, with edge features being weighed by the attention weights before being summed, and the resulting features are passed through an MLP to predict output edge existence likelihoods.

Our model is inspired by the the edge evolution framework in NEC-DGT [30], which operates directly on a fully connected graph topology. We found such unweighted message passing over all edges to be sub-optimal for scaling to the large number of objects in a house, potentially due to

---

[1]Implementation shared at https://github.com/Maithili/SpatioTemporalObjectTracking

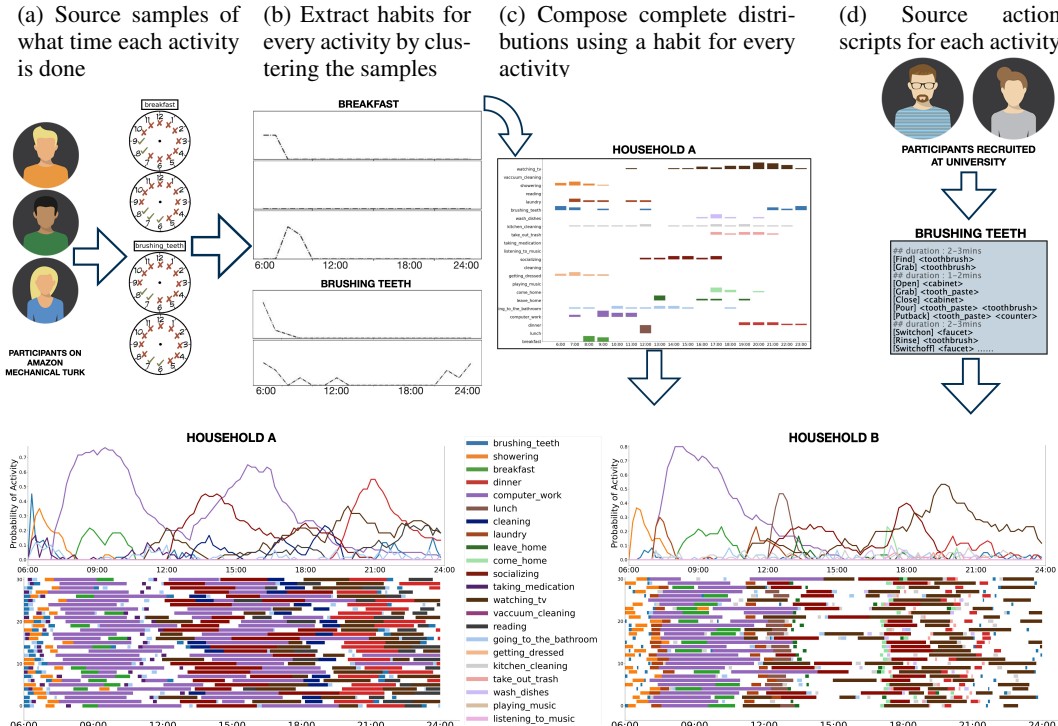

(a) Source samples of what time each activity is done

(b) Extract habits for every activity by clustering the samples

(c) Compose complete distributions using a habit for every activity

(d) Source action scripts for each activity

(e) Comparison of activity distributions and schedule samples of two personas shows common activities such as showering and breakfast, as well as activities unique to each household, for instance in Household A the user does cleaning and watches TV in the evening, and in Household B the user leaves the house after lunch and does laundry in the morning.

Figure 3: Process for generating the HOMER behavioral dataset.

over-squashing, so we employed the above attention mechanism. By predicting attention weights based on the sparse input graph topology, we allow the signal from those edges to be stronger. To avoid limiting the model's receptive field to existing neighbors, we allow all edges to contribute in the second step of aggregation, expecting the learned weights to emphasize important neighbors. Empirical results shown in Sec.7.3 confirm that such a mechanism is indeed beneficial.

## 5   Behavioral Simulation Dataset

To train and validate spatio-temporal object tracking for proactive open world robot assistance, we require a dataset capturing the locations of household objects as they move over the course of routine day-to-day human activities. Specifically, we require data that is longitudinal (spanning weeks or months) and captures naturally occurring variations in human behavior. Prior work has extensively documented routine activities in the home, particularly through datasets for activity recognition [36, 37, 38, 39]. However, existing datasets that contain object location data span only a few hours of data [40, 41, 42], and longitudinal datasets lack object location data [37, 43, 44].

We introduce the Household Object Movements from Everyday Routines (HOMER) dataset composed of routine behaviors spanning several weeks generated for five households representing habits of different individuals. The dataset is based on an apartment setting, where each apartment consists of four rooms and contains 108 objects. We used the VirtualHome simulator [45] to model human behavior, as it supports human agents, object interaction, and 33 high-level semantic command, such as *find*, *walk*, *grab*, without requiring low-level motion control. Figure 3 presents an overview of the dataset generation process. First, we obtained a list of activities of daily living relevant to in-home routines from the activity recognition literature [46]. Figure 3e shows the complete set of activities. We then separately crowd-sourced high level activity schedules and the low level action sequences to perform each activity, and used the resulting data to sample the daily routines.

**Activity Schedules**: To obtain realistic activity schedules, we collected data from 21 workers[2] on Amazon Mechanical Turk about which hours on a typical day from 6am to 12am they are likely to

---

[2]4F/17M, nearly equal participation from age groups of 25-35, 35-45, and over 45. Four of original 25 workers were omitted due to nonsensical answers, leaving 21 workers.

be doing each activity, Figure 3a. To build a probabilistic model that realistically captures variations in human behavior, we use clustering to extract diverse underlying habits of how each activity is performed, Figure 3b, and define a household using a combination of such habits for every activity, Figure 3c (see appendix for more details).

**Action Sequences**: In parallel to the above, we used the VirtualHome simulator to collect execution scripts emulating each activity of daily living, detailing the step-by-step action sequences, as well as the minimum and maximum time duration needed to do each action as shown in Figure 3d. We sourced this data from 23 participants, resulting in a set of action sequences, along with time duration ranges needed to perform each action, for all activities.

**Schedule Sampling**: Finally, using both the activity schedules and action sequences together, we generated the HOMER dataset capturing how objects move throughout the environment over time. We characterized each of our five households by a temporal activity distribution and an action script per activity, and used Monte Carlo sampling to generate samples of their daily routines[3], which are visualized in Figure 3e. When executed on the simulator, these routines provide a sequence of object arrangements, from which we derive states $X_t$ to train and evaluate our model.

The sample routines generated by our dataset emulate natural stochasticity in human routines, thus providing a realistic and challenging benchmark for a predictive system such as ours. Figure 3e depicts two example schedules from our dataset, which represent the distribution of activities in two different households. While day-to-day activity patterns in each household differ in timing, duration, order, and frequency, strong commonalities in behavior occur within each household. The user in household A tends to have breakfast only about half of the days and on some days they do so later, after using their computer for a little while. They also take breaks from computer work through the day to read, watch TV and socialize. They tend to work on the computer in the morning followed by taking a break around noon, similar to Household B, but then they skip lunch and continue working, whereas the user in Household B most often leaves the house after lunch. Other variations include allocating different times of the day for longer chores, such as Household A takes time to do cleaning towards the evening, whereas Household B does laundry earlier in the day. Thus, we create a challenging test bed for our models, as they need to make good predictions to capitalize on the opportunity to assist the user, but also avoid making incorrect predictions, particularly for hard-to-predict activities, which might cause disruptions to the user's natural routine.

## 6    Evaluation

We evaluate our predictive model on sequential scenes from each of our five household datasets and report results averaged across all five. We use $\tau_i$ of 1 day, 12 hours, 6 hours, 3 hours, 1 hour, 30 mins, and 10 mins, construct each of the our MLPs with a single hidden layer consisting of 20 neurons and ReLU activation, and use Adam optimizer with a learning rate of $10^{-3}$. Starting with a known graph $G_t$ at time $t$, we use our network to predict the probabilistic graph one step (10 mins) into the future $p(G_{t+1})$, and feed it back into the model iteratively to predict further into the future. Finally we derive the desired posterior estimate $\hat{G}_{t+\delta}$.

### 6.1    Baselines

We compare our predictive model against two prior works on open world object tracking:

**Static Semantic**, adapted from [16], calculates static priors over object locations using observed object-location relation frequencies, and uses these priors to update the estimate on every step. To adapt the noise model to our topological formulation, we use a tunable probability of change and spread belief uniformly over all potential locations, as opposed to nearby areas in metric space.

**FreMEn**, adapted from [17], uses past experience to model the prior probability of existence of object-location relationships as periodic functions in time. We maintain beliefs over topological relations instead of the metric occupancy grid used in [17]. The final belief is a combination of the given state and periodic temporal priors, with a tunable time-decaying weight as suggested in [17].

### 6.2    Metrics

Given the current state $X_t$, represented by the graph $G_t$, our objective is to predict the state graph $\delta$ timesteps in the future, $\hat{G}_{t+\delta}$, and use it to infer object relocations, $r(o_i, l_1, l_2)$ caused by human

---

[3]The complete dataset can be found at https://github.com/GT-RAIL/rail_tasksim/tree/homer/routines, and further implementation details are available in the appendix.

actions, signifying the movement of object $o_i$ from its original location $l_1$ to destination $l_2$. The set of such relocations, $\hat{\mathcal{R}}_{t:t+1}$, predicted to happen one step into the future $t+1$ can be written as

$$\hat{\mathcal{R}}_{t:t+1} = \{r(o_i, l_1, l_2)|e_{i,l_1} \in \hat{G}_t, e_{i,l_2} \in \hat{G}_{t+1}, l_1 \neq l_2\}$$

To assist $\delta$-steps into the future, the robot needs to predict relocations $\hat{\mathcal{R}}_{t:t+\delta}$ obtained from sequential predictions from time $t$ to $t+\delta$ and reconciling multiple relocations:

$$\hat{\mathcal{R}}_{t:t+\delta} = \hat{\mathcal{R}}_{t:t+\delta-1} \cup \{r(o_i, l_1, l_2)|r(o_i, l_1, l_2) \in \hat{\mathcal{R}}_{t+\delta-1:t+\delta}, r(o_i, l_1, l_3) \notin \hat{\mathcal{R}}_{t:t+\delta-1}\}$$

In addition, we can extract the set of objects that are relocated as a part of relocations $R$ as

$$\mathcal{O}(\mathcal{R}) = \{o_i|r(o_i, l_1, l_2) \in \mathcal{R}\}$$

We evaluate model selected relocations against the ground truth relocations that would have been made by the human user in the absence of any assistance ($\mathcal{R}_{t:t+\delta}$). The exact order and precise timing of the robot's relocations is not critical as long as they occur before the human-generated event (e.g, the bowl or cereal could be taken out first). Hence, we measure the predictive performance of relocations over the entire proactivity $\delta$.

For our evaluations, we separately consider objects that were used by the human user between time $t$ and $t+\delta$, $\mathcal{O}(R_{t:t+\delta})$, and the remaining unused objects. While moving the unused objects might not cause a direct disruption in the user's routine, it would be disconcerting if objects randomly moved in the house. Hence, we want to maximize the fraction of unused objects that are left in their places ($\{o_i|o_i \notin \mathcal{O}(R_{t:t+\delta}), o_i \notin \mathcal{O}(\hat{R}_{t:t+\delta})\}$). The more interesting cases are the objects that were used by the human user. Robot's predictions for these objects can fall into one of three categories. First, objects whose predicted movements are **correct** ($\mathcal{O}(\hat{\mathcal{R}}_{t:t+\delta} \cap \mathcal{R}_{t:t+\delta})$). Second, objects that are predicted to move to the **wrong** locations (($\mathcal{O}(\hat{\mathcal{R}}_{t:t+\delta}) \cap \mathcal{O}(\mathcal{R}_{t:t+\delta})) \setminus \mathcal{O}(\hat{\mathcal{R}}_{t:t+\delta} \cap \mathcal{R}_{t:t+\delta})$); these are especially undesirable as the robot would move objects to a random place, making it harder for the user to find them. Third, objects that were **missed** ($\mathcal{O}(\mathcal{R}_{t:t+\delta})) \setminus \mathcal{O}(\hat{\mathcal{R}}_{t:t+\delta})$) and hence were left untouched, which is still undesirable but the user would presumably easily find them in their original locations. Of these three categories, we want to maximize the correct relocations, while minimizing the wrong relocations on the objects that the user uses through their routine.

# 7 Results

Below, we compare the performance of our model against baselines, investigate the effect of fewer observation days for training, show the importance of our attention mechanism and time encoding, and finally validate our model on a real robot. Our quantitative comparisons are based on 50 days of training data and a 30 minute proactivity window, however we show results over varying proactivity windows in Section 7.1, and discuss the effect of fewer training observations in Section 7.2.

## 7.1 Performance comparison against Baselines

Our model moves 11.1% more objects to correct locations and 11.5% fewer objects to wrong locations out of those used by the user, compared to FreMEn, our leading baseline. Of the objects not used by the user, our model mistakenly moves only 0.8% compared to over 4% objects moved by our baselines. These results are summarized in Table 1. For these results, we train each model separately on each household, and test on a withheld set of 10 days from the same household distribution.

| Method | % Used Objects | | | % Unused Objects | |
|---|---|---|---|---|---|
| | Correct | Wrong | Missed | Correct | Wrong |
| Static Semantic | 23.04 | 16.16 | 60.80 | 91.06 | 8.94 |
| FreMEn | 29.42 | 15.28 | 55.29 | 95.30 | 4.70 |
| Ours | **40.51** | **3.79** | 55.70 | **99.19** | 0.81 |

Table 1: Comparison of predictive performance on used and unused objects for a 30 minute proactivity window of our method against baselines

Performance over used and unused objects are shown in Figure 4 for a predictive window of 10 minutes (leftmost bar) to 120 minutes (rightmost bar) with intervals of 10. More objects are used for longer proactivity windows, as can be seen by the total height of each bar representing the total number of used or unused objects in that window. On increasing the proactivity window upto 40 minutes (the first four bars), our method as well as FreMEn predict a larger fraction of moved objects correctly, because a larger proactivity window allows for some error in predicting the time of a future event. However beyond the 40 minutes, the performance starts dropping owing to the natural uncertainty in predicting farther out in the future.

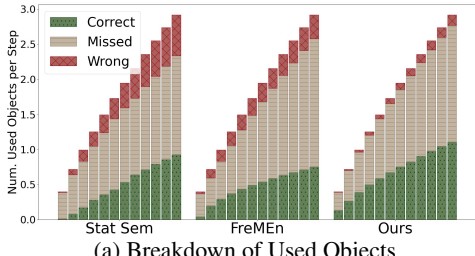
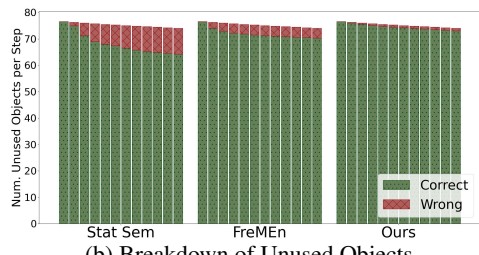

(a) Breakdown of Used Objects        (b) Breakdown of Unused Objects

Figure 4: Our methods moves the most used objects to correct places and fewest used objects to the wrong places for proactivity $\delta$ of 10 minutes (leftmost bar) to 120 minutes(rightmost bar), at intervals of 10. Moreover, our method touches fewer unused objects than our baselines.

Across different proactivity windows we observe that FreMEn makes a lot of wrong relocation predictions on both used and unused objects. This is because FreMEn rearrangements are based on a prior that is conditioned entirely on *time*, without considering current environment state $X_t$. As a result, natural deviations in user behavior (e.g., early breakfast) will cause the model to place objects in their typical locations (e.g., bowl back in the cabinet) even when it should remain unmoved while still in use, or when they should be moved to the next location (e.g., bowl moved to sink). By contrast, our model's ability to model typical state progressions, allows it to handle such situations (e.g. bowl moves to the sink after coming out). The Static Semantic baseline demonstrates weaknesses similar to FreMEn because its prior is agnostic to both $X_t$ and time. In a real world scenario, both baseline methods would cause the robot to constantly rearrange objects in the house, often misplacing objects that would be needed and moving unrelated items.

Our model robustly predicts routine object movements, while maintaining a conservative behavior over non-routine movements. In Household B (Figure 3e) the user takes a shower routinely and consistently, so our model is able to reliably predict associated movement of the towel. In situations that are especially challenging to predict (e.g., highly inconsistent breakfast routine in household A), our model acts conservatively and avoids moving objects, resulting in fewer wrong movements (3.79%). A consequence of the model's conservative nature is that when the user does have breakfast, no proactive assistance is provided. This results in missed assistive opportunities (55.7%), but we believe real world users would prefer to avoid having objects moved unnecessarily. Sometimes our model fails to predict assistive actions for activities with a large time variation, such as relocating the TV remote for household B (Figure 3e), despite them watching TV almost everyday.

### 7.2 Training Data Efficiency

We evaluate data efficiency by training each model using varying amounts of training data, from 5 to 50 days of observations. As shown in Figure 5, our method improves performance as more data becomes available, and outperforms both baselines with as little as 5 days of observations. The relatively simpler representation of the baseline methods enables those models to achieve high performance sooner, but hurt their improvement with more data as they cannot utilize it to learn more complex patterns. For instance, FreMEn independently models each object being in each location, and cannot learn correlations between different objects. Our model can leverage such correlations (e.g., cereal and milk movements being correlated).

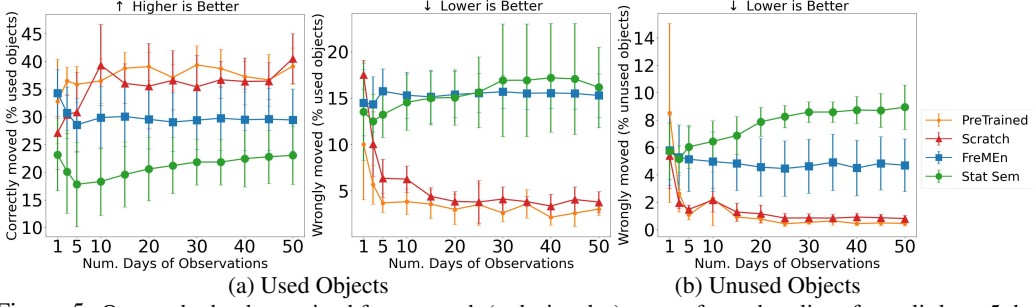

(a) Used Objects        (b) Unused Objects

Figure 5: Our method, when trained from scratch (red triangles), outperforms baselines for as little as 5 days of observations, and shows improved performance when pre-trained (orange dots) on other households.

Additionally, we evaluate the benefits of cross-training across households to take advantage of commonalities in human behavior. We use data from four households to pre-train a model, and fine-tune and test on the train and test splits, respectively, on a fifth household. Compared to a model trained

from scratch, the pre-trained model starts off with higher performance over the used objects (Figure 5a), but slightly lower performance over unused objects (Figure 5b) as it imposes other households' patterns on the new household. As more data is made available, the performance gap on % correct predictions closes, but the pre-trained model makes slightly fewer % wrong predictions on both used and unused objects, implying that training on more varied data provides a good initialization. By comparison, for both baselines cross-training across households reduces performance.

### 7.3 Model Analysis

We perform an analysis to measure the impact of our attention mechanism and time representation. The purpose of our attention mechanism is to avoid over-squashing caused by message passing over a fully connected graph topology with a large branching factor. Empirical results over used objects shown in Table 2 prove its benefit as our model outperforms a version without such a mechanism. Our model also exhibits superior performance over using a linear representation of time as a single number in minutes.

| Method | % Correct | % Wrong |
|---|---|---|
| Ours | **40.51** | **3.79** |
| Without Attention | 25.85 | 8.20 |
| Without Time Enc. | 24.84 | 8.29 |

Table 2: Model Analysis

### 7.4 Robot Validation

Finally, we present a proof-of-concept demo of our system on a Fetch robot assisting with a breakfast routine, as shown in Figure 6 and detailed in our accompanying video. The shown scenario demonstrates anticipatory assistance through taking out of cereal, milk and bowl prior to the user's breakfast. Additionally, the robot anticipates the need to clean up, resulting in the restorative actions of placing the cereal box and milk back, and leaving the bowl in the sink. One benefit of our object-based proactive assistance model is that it is robust to certain variations in the user's routine. When the user skips breakfast, even though the robot prepared the breakfast items, it will also replace them once breakfast time has elapsed.

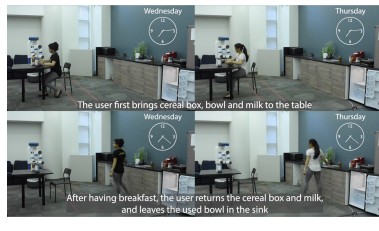 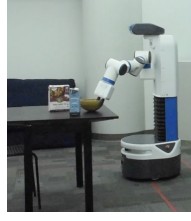 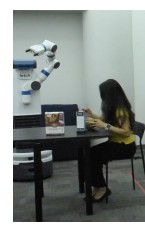 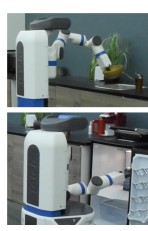 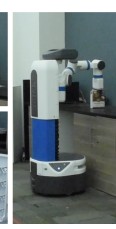

(a)             (b)             (c)             (d)

Figure 6: The robot observes user behavior and learns patterns therein (a). The robot uses the learned model for proactive assistance (b), reducing effort to the user (c) both before the breakfast activity, and after (d).

## 8 Discussion, Limitations and Future Work

In this paper, we introduce an object-centric perspective to providing proactive robot assistance, present a novel dataset for longitudinal object tracking, and contribute a novel generative graph neural network formulation for predicting future environmental scenes. Our results show that our model significantly outperforms two prior techniques for modeling environmental dynamics, even if given as little as 5 days of training data. We additionally demonstrate that the model can be used on a real robot to accurately predict future objects movements and plan anticipatory and restorative actions that benefit the user, without the need for activity recognition and subsequent analysis of used objects in each activity. Such assistance does not require explicit user commands, reducing the burden on the user.

**Limitations and Future Work:** As discussed earlier, leveraging only object data removes reliance on potentially incomplete or inaccurate activity recognition models, which can be a strength in complex real-world deployment scenarios. However, the lack of activity modeling is also a limitation because the robot is unaware of the user's current actions. For example, if the user is taking an unusually long time to eat breakfast, the robot may clean up too early. Future work should combine the capabilities of both object-based and activity-based systems. An additional limitation is that our method does not currently account for object state. Future work should expand the representation to condition object placement on state information (e.g., place clean bowls in cabinet and dirty bowls in the sink). Finally, human behavior is rarely deterministic and user preferences may be hard to predict (e.g., cereal or yogurt for breakfast). Future work should integrate user interaction to enable the robot to confirm with the user before performing actions with high uncertainty, and include a user study to understand how users react to such a system.

**Acknowledgments**

This work is sponsored in part by NSF IIS 2112633.

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
