# OpenReview forum: "Proactive Robot Assistance via Spatio-Temporal Object Modeling"
_robot-learning.org/CoRL/2022/Conference — CoRL 2022 Poster_

### Official Review · Reviewer_KdUK · 2022-07-26

**Originality:** Very Good
**Technical Quality:** Very Good
**Clarity Of Presentation:** Excellent
**Impact:** 4

**Recommendation:**

Strong Accept: I recommend accepting the paper and will argue for my recommendation even if other reviewers hold a different opinion.

**Summary:**

This work provides a solution to the problem of robotic proactive assistance in everyday human routines. The paper introduces a novel method for successfully solving the tasks at hand. The proposed method is a generative graph neural network that utilises attention and time indexing and provides a useful and fairly large-scale behavioural simulation dataset. This paper was pleasant to read and my opinion is overall positive.

**Issues:**

### Presentation
There are not many issues with this work in my opinion. The paper may become easier to parse if some of the figures were made a bit less complex. For example, Figure 1 can be simplified to just illustrate the idea. Right now it seems overly cluttered which makes it a challenge (at least for me) to grasp the idea from looking at it. Figure 2 could also be made clearer. Right now it is hard to tell what are the individual components (e.g. which part is the attention, where is the time indexing) what are the different network sizes. The figure is also a bit too small.

### Literature
[i] this work seems relevant in the context of collaborative assistance so it may be worth citing.

[i] Penkov, S., Bordallo, A. and Ramamoorthy, S., 2017, May. Physical symbol grounding and instance learning through demonstration and eye tracking. In 2017 IEEE International Conference on Robotics and Automation (ICRA) (pp. 5921-5928). IEEE.

**Quality Of The Limitations Section:**

Limitations are addressed clearly

**Reviewer Expertise:**

3: The reviewer is fairly confident that the evaluation is correct

**Robotics Focus:**

Sufficient demonstration on hardware

**Strengths And Weaknesses:**

### Strengths
- novel solution to an important problem
- clearly written and well evaluated work
- a large scale dataset

### Weaknesses
- Depends on hand-crafted features
- The method does not explicitly take into account the user

**Summary Of Recommendation:**

Although I am not an expert on proactive robotic assistance I think this paper should be accepted to the conference. It provides a **clearly written** and very **well evaluated novel solution** to a valuable set of problems. In addition, it provides a seemingly **challenging relatively large-scale dataset** that can be instrumental for future research on the problem.

---

> ### Author Response · Authors · 2022-08-25
> **Response to review by Reviewer KdUK**
>
> We appreciate the comments and thank the reviewer for the positive review, and for bringing this work to our attention. We agree that the first two figures might appear too busy. We will revise them to be clearer and annotate them to better identify the individual components.

---

### Official Review · Reviewer_fzEn · 2022-07-29

**Originality:** Good
**Technical Quality:** Good
**Clarity Of Presentation:** Very Good
**Impact:** 3

**Recommendation:**

Weak Accept: I recommend accepting the paper, but will not argue for my recommendation if the majority of other reviewers have a different opinion.

**Summary:**

The authors present a model of object spatio-temporal dynamics and show how it can be used in the context of proactive robotic object relocation. They model the state of objects in an environment with a graph and learn a model that predicts how the graph will evolve over time based on a novel simulated dataset of longitudinal object relocation activities in homes. A robot can then leverage this model to predict that some objects would be needed (e.g. a bowl in the morning around breakfast time) and proactively relocate them.

**Issues:**

* Provide discussion of the qualitative performance of the approach. What kinds of errors does the model make? What are plausible paths to addressing these deficiencies?

* Is a limitation that the robot needs to have a full grounding of the graph at any instead before predicting forward? How feasible would it be for future work to relax this need?

* Minor: The small number of users sampled for their habit information struck me as odd. Some mention of why the size of the sample was selected would be helpful. Were there existing, alternative sources for this kind of information? It seems like the kind of thing that would be surveyed or collected in diary studies for any number of purposes, and likely would’ve come from more representative sampling methods.


**Quality Of The Limitations Section:**

Additional details required

**Reviewer Expertise:**

3: The reviewer is fairly confident that the evaluation is correct

**Robotics Focus:**

Sufficient demonstration on hardware

**Strengths And Weaknesses:**

Strengths

* A pillar of the author’s approach is that an object-centric model (which does not explicitly represent agents in the environment at all) can be useful for predicting when an object should be relocated proactively. This is an interesting contrast with most work that tries to make robots proactive, which generally have some model of what a human is doing and generate interventions that would assist with the speculated action. Models of people are hard, so I think exploring other routes is interesting.

* The dataset and the method the authors used to collect it are interesting. It’s hard to collect longitudinal data on human activities, especially if you care about timescales like weeks or months. The way in which the authors interleaved the collection of coarse distributional information about habits, then used this to make personas which guide fine grained activity data collection in simulation is a good way to circumvent this problem.

* The quantitative evaluation is good. Comparison to baselines is appreciated. The performance of the baselines is also well connected to the limitations of their respective models.

Weaknesses

* Limited qualitative evaluation of the proposed model. What kinds of mistakes does the model make? When the wrong relocation is predicted, is it more often the object, the destination, or the time that is wrong? How severe are mispredictions? Do objects end up put in weird places? Do they usually get returned later? There’s a good discussion of how the baselines fail on lines 228-237, but no similar discussion for the proposed approach.

* The title of the paper, “Proactive Robot Assistance” may prime users to expect some of the more common forms of assistance from the industrial human-robot collaboration literature like pre-kitting or changing the state of objects with tools (like tightening bolts). It’s not obvious how the approach could provide “assistance” in a general sense like this. The authors study object relocation, which is one (valuable!) type of assistance.

* The absolute level of performance is low. While the same can be said for many robot learning endeavors, here it works against the core argument that you can provide useful proactive assistance with a spatio-temporal object model. 70% of the times the robot chooses to move something, the robot relocates something that didn’t need to be relocated, relocates something to the wrong place, or both. Regardless of whether the approach beats other object dynamics models, this would seem to indicate that the difficulty of predicting even just the consequences of human actions (relocation) is high, and unlike approaches that predict interventions based on human activities, this method doesn’t afford natural interactions where the robot can ask if it should do something in cases of uncertainty. Something like “It looks like you might be getting ready for bed. Should I go ahead and lay out your night clothes?” is easy if you are detecting activities as the basis for predicting interventions. But here, you might just ask, “should I move a glass onto the counter?” around 9PM. Depending on the true context that the user is in, this could seem like a nonsensical question.

**Summary Of Recommendation:**

The authors show an interesting take on how to provide proactive assistance, but given the limitations of the framing, and given the robotics audience for a paper like this, a deeper discussion of the performance is warranted. A strength of the work is in highlighting that, fundamentally, there's no reason why a robot couldn't realize some basic, useful assistance with a relatively constrained understanding of the world. But the paper seems to gloss over some of the practicalities that come with the approach in its current state.

**Update after rebuttal period:**

Your response has alleviated the bulk of my concerns. I remain concerned about whether this formulation of assistance is ultimately desirable or workable. I agree that making the system more conservative is a reasonable path forward, but the path towards natural interaction to clarify or gather corrections is inherently challenging in this formulation. I agree that this isn't within the reasonable scope of this work. It is also, as another reviewer pointed, an open question whether a system that completely satisfied the objective would actually be liked by users. There is some risk that future work may focus on improving performance while leaving these substantially more important questions unresolved

I think there will be interested readers for this work, so I am updating my rating accordingly.

---

> ### Author Response · Authors · 2022-08-25
> **Response to review by Reviewer fzEn**
>
> We thank the reviewer for their detailed feedback and address individual comments below.
>
> **“Provide discussion of the qualitative performance”**
>
> We agree that it is necessary to qualitatively understand a system, especially when it fails. Below are several examples, which we will include in the results discussion.
> - The persona A (figure 3e in the paper) misses breakfast on 21 of the 50 training days, and has breakfast quite late on 5 of the remaining 29 days. Due to the lack of a consistent routine, our model responds conservatively by not predicting cereal, bowl and milk to come out, causing a significant drop in recall and destination accuracy. However, it does learn to put them away after some time once they are out.
> - On the flip side, when a person skips doing what is otherwise a routine activity, the robot wrongly predicts object movements, causing a reduction in precision. For instance, persona B washes dishes most evenings, but when the user skips this activity or does it later, bringing out the dish towel will be a wrong prediction. Our method achieving a good precision indicates that this type of errors are relatively less prevalent.
> - Finally, failed opportunities may occur, such as for persona B (figure 3e), the robot often fails to relocate the TV remote despite them watching TV almost everyday. This might be due to the large variation in the time of when the person watches TV.
>
> **“The absolute level of performance is low”**
>
> - **Clarification of results** The metrics we reported may have been too abstract, hiding some of the behavior of the system. Here we clarify the behavior and report an updated result:
>   - **Clarification**: When we reported a destination accuracy of x%, this implied that (1-x)% of the time one of two types of errors occurred – either an object was never relocated, or it was relocated, but to the wrong location. To improve clarity, we will now report these values independently. Specifically, our results are that 38% of objects were relocated correctly, only 4% were relocated to the wrong place, and 58% of the time objects that should have been moved are left in their place. This shows that our model is somewhat conservative about predicting object movements.
>   - **Note**: Our performance would be artificially higher if we only considered routine object movements. Instead our dataset includes ALL household objects – both routine and non-routine objects – for a more realistic setting, which makes the prediction performance lower.
>   - **Overall task difficulty**: Overall, the prediction task is difficult. A close comparison can be made to visual activity anticipation, where activity labels are predicted a few frames ahead in a video. This is an established field and yet SOTA methods still hover around ~30-40% accuracy [Wu 2022].
>   - We **fixed a bug** in our simulator that slightly improved our performance, raising destination accuracy from 32% to 38%. The Virtual Home simulator includes an avatar that walks through the space to move objects. While we had removed the avatar, we had failed to exclude its clothes, which continued to teleport across the rooms and confuse our model.
>
> - **Improving precision**. To mitigate errors pointed out by the reviewer (relocating the wrong object or relocating an object to the wrong place), we added a tunable parameter representing the confidence threshold for executing an action. We present results in the attached pdf and will include them in our revised paper.
>
> **“What are plausible paths to addressing these deficiencies?”**
>
> To reduce the amount of disruptive false positive actions, the system can be made more conservative by tuning the confidence threshold mentioned above, or restricting the system to a set of actions that are deemed okay by the user.
>
> As we indicate in the limitations section, user interaction using time and other object locations as context can be leveraged to reduce mistakes, but a detailed study on interaction generation is outside the scope of this paper.
>
> **“Is a limitation that the robot needs to have a full grounding of the graph…?”**
>
> No, this is not a limitation, our model is capable of using probabilistic inputs. For our multi-step evaluations, we iteratively feed the probabilistic outputs back into the model, and it is able to work well with such inputs. For continuous deployment, one could use classical filtering approaches to maintain belief over the state from available observations, and use that as input to the model.
>
> **“The title of the paper, “Proactive Robot Assistance” may prime users to expect some of the more common forms of assistance…”**
>
> We appreciate the feedback and will make appropriate adjustments to clarify that the assistance is with respect to fetching objects.
>
>
> [Wu 2022]  Wu, Chao-Yuan, et al. "Memvit: Memory-augmented multiscale vision transformer for efficient long-term video recognition." Proceedings of the IEEE/CVF Conference on Computer Vision and Pattern Recognition. 2022.

---

### Official Review · Reviewer_EX28 · 2022-07-31

**Originality:** Fair
**Technical Quality:** Good
**Clarity Of Presentation:** Good
**Impact:** 3

**Recommendation:**

Weak Reject: I recommend rejecting the paper, but will not argue for my recommendation if the majority of other reviewers have a different opinion.

**Summary:**

This work focuses on proactive robot assistance for daily/routine tasks in a home environment, e.g., bringing the breakfast materials on to the table. This assistance relies on an object-centric prediction pipeline, based on a graph neural network model. With such an object-centric focus, the dataset provided also differs from the existing ones that are mainly based on activity recognition aspect.
The data covers a diverse set of activities while spanning a long-horizon (several weeks) timeframe. Human behavioral patterns are segregated into five personas to capture commonalities in the activity patterns. The evaluations consider comparisons to two baselines. The proposed approach achieves a better performance on several metrics.

**Issues:**

- a better explanation on how/why different personas are selected/clustered.
- some evaluations on assistance with a finer time window
- any possibility to compare against activity-recognition based approaches?
- time assignments for action sequences: how reliable/accurate they are?
- what do you mean by 'combined the activity schedules and action sequences' when they come from two different set of people/environment/setup?
- clarification on the 'type' of assistance that the proposed method aims for... be more precise in the title and in general within the main text.
- why do you train your model per persona? how do you decide which model to use during testing (e.g., if the persona is unclear)?


**Quality Of The Limitations Section:**

Additional details required

**Reviewer Expertise:**

4: The reviewer is confident but not absolutely certain that the evaluation is correct

**Robotics Focus:**

Relevant but unlikely to deploy to hardware in near future

**Strengths And Weaknesses:**

strengths:
- a different take on proactive assistance by focusing on object-centric modeling showcasing the pros and cons of such an approach
- better performance on a few metrics compared to a couple of other approaches
- a new dataset containing information on spatio-temporal dynamics of household objects

weaknesses:
- no methodological novelty
- without explicit activity recognition, the predictive model is ignorant of the activity and object states, possibly resulting in inaccurate/unhelpful/redundant assistance
- human behavioral patterns/personas are explicitly (pre-)identified, i.e., independent of the predictive model architecture -> possibly limits the application of the proposed GNN architecture out-of-the-box
- assistance time-window seems to be too coarse (10 minutes), i.e., for tasks that (might) require quicker assistance cannot be tackled out-of-the-box -> diminishing the value proposition of the approach
- the data comes from a simulator, and it's unclear how well it captures daily routines of people, especially in terms of finer time scales and variability
- not much applicability in a real setting, because it still acts as an activity/time-series recognizer/predictor without much focus on robot awareness / autonomy / assistance


**Summary Of Recommendation:**

The paper offers a different perspective on proactive robot assistance by focusing on how and when objects are interacted with during routine tasks. Considering its weaknesses (some already acknowledged by the authors), it might turn out to be a good stepping stone for future work, especially if/when the model can also capture the 'true state' of the environment/activity. However, at its current state, I do not think that it provides a significant contribution in robot assistance tasks. Even though a superior performance is highlighted compared to two baselines, these baselines also consider object-tracking based predictions, and it's unclear how this method compares against a more traditional activity recognition based approach.

---

> ### Author Response · Authors · 2022-08-25
> **Response to review by Reviewer EX28**
>
> We thank the reviewer for their detailed feedback. We understand the reviewer has concerns about the realism and detail of data, and some confusion on how the data is generated. We have attached a document outlining and clarifying the characteristics of our dataset. We address other comments below.
>
> **“assistance time-window seems to be too coarse (10 minutes), i.e., for tasks that (might) require quicker assistance cannot be tackled out-of-the-box”**
>
> The reviewer’s vision of near real-time proactive assistance is compelling and one that we strive towards. The 10-minute interval is an adjustable parameter and can be easily changed by sampling from the model more frequently. We chose this interval for a few reasons:
> - Each time the model predicts the new state of the world, multiple objects may have moved (e.g., cereal and milk and bowl), so the time step does not limit the level of assistance the robot can provide in terms of the number of objects moved.
> - We are exploring assistance at the full house scale, which requires mobile manipulation. The current state of mobile manipulation limits how quickly the robot is able to complete actions. Prior work that has occurred at smaller time scales for more real-time assistance has all focused only on tabletop scenarios, such as the toy cooking scenario used in [Oh 2021], which are limited to manipulation only and afford clear activity observations within a single camera field of view. We expect our actions to take longer as the robot has to navigate to new locations, possibly open drawers, etc.
>
> **“any possibility to compare against activity-recognition based approaches?”**
>
> There are several reasons we do not include this comparison. First, existing activity and object centric methods are aimed at shorter time scales, and leverage different types of information which are available at those scales. Second, activity recognition approaches depend on observing the user at all times, which is likely to be expensive and potentially invasive of privacy over longer time scales. Finally, we cannot assume user actions to always be clearly visible to the robot across the household (multiple rooms), and without longitudinal datasets containing sensory data pertaining to activities we cannot model the performance of an activity recognition system in a realistic way.
>
> **“without explicit activity recognition, the predictive model is ignorant of the activity and object states, possibly resulting in inaccurate/unhelpful/redundant assistance”**
>
> We agree that activity observations hold useful information for the task of proactive assistance, and that such information is complementary to our system. In this work, we are interested in understanding how far assistance can be taken without reliance on activity recognition (which can be quite noisy in the wild). Exploring synergies between these capabilities is an important direction for future work. We are particularly interested in the future to extend this work to consider limited and highly noisy activity recognition systems that opportunistically or sparsely observe the user.
>
> **“human behavioral patterns/personas are explicitly (pre-)identified, i.e., independent of the predictive model architecture”**
>
> We disagree with (or possibly misunderstood) this comment.
>
> Yes, it is correct that behavioral patterns / personas are pre-identified. We view each persona as data from a different “household” in which the house occupant has their own routines and habits.
>
> Yes, our model architecture is not dependent on any given persona. We view this as a strength rather than limitation as our results show that our model applies across all the various personas we tested.
>
> **“why do you train your model per persona? how do you decide which model to use during testing”**
>
> We view a “persona” as equivalent to a “household” and capture the occupant’s routines (we will revise our terminology since our initial wording appears to cause confusion). We envision a robot deployed in a household to have its own model of the occupant’s routine. Accordingly, we originally trained and tested a model on data from a single persona.
>
> In response to Reviewer QKuy, we ran additional experiments with results suggesting that pre-training on multiple personas further improves performance (see other review response for more details) and we will be adding these results to the paper.
>
> **“clarification on the 'type' of assistance that the proposed method aims for...”**
>
> We appreciate the feedback and have made appropriate adjustments to clarify that the assistance is with respect to fetching objects.
>
>
> [Oh 2021] Oh, Nayoung, et al. "A robot capable of proactive assistance through handovers for sequential tasks." 2021 18th International Conference on Ubiquitous Robots (UR). IEEE, 2021.
> [Guo 2019] Guo, Xiaojie, et al. "Deep multi-attributed graph translation with node-edge co-evolution." 2019 IEEE International Conference on Data Mining (ICDM). IEEE, 2019.

---

### Official Review · Reviewer_QKuy · 2022-08-05

**Originality:** Fair
**Technical Quality:** Good
**Clarity Of Presentation:** Very Good
**Impact:** 3

**Recommendation:**

Weak Reject: I recommend rejecting the paper, but will not argue for my recommendation if the majority of other reviewers have a different opinion.

**Summary:**

The paper presents a graph prediction method designed to predict how object move in a household as a result of a person's daily activities. The paper also presents a novel simulated dataset for this problem. The method outperforms naive baselines. A demonstration with a robot shows that a robot could in principle use this method to anticipate where objects may be needed in the future.

**Issues:**


The main issue that I see is with the problem formulation itself. Why not learn from multiple households and people at the same time? Or at least formulate the problem in a way that a learned model from one household/person can be adapted to a new household/person through, let's say, domain adaptation?

Can you also elaborate on what you mean by "open-world"? That phrase appears a few times and it has a very precise meaning in the AI literature, e.g., worlds in which things change (e.g., all of a sudden the human's behavior pattern changes, or new objects appear, or a new set of location is added to the household, etc.). To me, it doesn't appear that the simulated environment is actually an open world as defined in the AI literature.

The baseline methods do not seem to improve as more training data is available. Any explanation why?

Finally, the  motivation needs to address the issue is what happens when the user loses some locus of control -- how would people actually react if objects in their house start moving without their knowledge?




**Quality Of The Limitations Section:**

Additional details required

**Reviewer Expertise:**

3: The reviewer is fairly confident that the evaluation is correct

**Robotics Focus:**

Relevant but unlikely to deploy to hardware in near future

**Strengths And Weaknesses:**


Strengths:

The paper is written well and the description of the problem and method are clear. The simulator used to generate the data is novel and potentially could be expanded into a paper on its own. A technical demonstration shows that in principle, a robot could use this method.

Weaknesses:

Results are only reported on simulated data. The baselines are rather naive while the proposed graph prediction method is fairly straightforward and very similar to other methods applied on different domains. A major limitation is that it is unclear how a learned model would transfer to a new household.


**Summary Of Recommendation:**


The recommendation is based on the fact that the graph prediction method in itself is routine while at the same time, the results are based on just simulated data.

---

> ### Author Response · Authors · 2022-08-25
> **Response to review by Reviewer QKuy**
>
> We thank the reviewer for their detailed feedback. Before addressing specific comments, we would like to reiterate our contributions. Our primary contributions are 1) the novel formulation of the proactive assistance problem towards long-horizon assistance through object relocations, 2) a proposed solution to that problem in the absence of access to an activity recognition system because deploying such systems in a home is likely to be expensive, potentially invasive of privacy, and difficult to train.
>
>
> **“A major limitation is that it is unclear how a learned model would transfer to a new household…. Why not learn from multiple households and people at the same time?”**
>
> We thank the reviewer for pointing out this missing information. We had focused on making our approach sample efficient (requiring only 5 days of training data) and tailored to data derived from a single household. However, since human behavior has commonalities across households, it is interesting to consider how the learned models can generalize across households.
>
> We ran experiments to assess how well a learned model can transfer to a new household and found that pre-trained models achieve higher precision. We have detailed these results in the attached pdf and will include them in Section 7 in our revised paper.
>
> **“Results are only reported on simulated data.”**
>
> Yes, we constructed a novel dataset based on crowdsourced human activity data to ensure that we could evaluate our approach in a longitudinal setting. We show a proof of concept demo on a robotic system. Collecting many days of observations from multiple households in the real world would be very exciting, but is currently beyond our practical scope.
>
> **“The baselines are rather naive…”**
>
> We are not aware of any prior work on predictive modeling of object movements towards assistance. As a result, we compare against other applications in robotics that require a similar underlying reasoning framework of object location tracking. If we are missing a specific baseline, we welcome feedback and suggestions for additional methods to test.
>
> **“The graph prediction method in itself is routine”**
>
> Since we are the first to solve this problem as formulated, we borrow a base architecture from other fields and make significant changes, which are responsible for nearly 20% of our performance, as shown in our ablations. Having adopted a generative GNN architecture[Guo 2019], we add a novel mechanism to avoid the fundamental issue arising from the amount of inductive bias in GNNs, where using a known sparse structure leads to high inductive bias, but ignoring it (using a fully connected model [Guo 2019]) loses precious structure. We find a balanced solution by utilizing the constraining structure to, in essence, predict a probabilistic graph structure (in the form of attention weights), which is not constrained entirely to the input graph, but uses information from that structure. We will more clearly highlight these technical details in the revised paper.
>
> **“how would people actually react if objects in their house start moving…?”**
>
> This is a very important topic, which we will add to the discussion in the limitation section of our paper; our current study does not shed light on this research question, and the user study required to sufficiently address this point is outside the scope of this paper.
>
> **“it doesn't appear that the simulated environment is actually an open world as defined in the AI literature”**
>
> Thank you for the correction, we agree that the simulated environment does not exhibit the same level of complexity in terms of changing human behavior and new objects and locations. We will avoid using this term.
>
> **“baseline methods do not seem to improve as more training data is available. Any explanation why?”**
>
> Fundamentally, the limited expressive abilities of the baselines hinder learning of more complex patterns, which our model can capture. Once they extract the information they can use from a few samples, the new samples contribute little to improve their performance. Specifically:
> - FreMEn learns a temporal pattern for each object being in each location independent of other objects, and cannot learn correlations between different objects. For instance, all models might fail to predict the cereal and milk coming out if the person doesn’t have breakfast very often, but when the person does bring cereal out, our model can leverage the correlation between cereal and milk and predict milk to come out, but FreMEn will fail to do so.
> - Static Semantic baseline does not take into consideration the time when calculating its prior. As a result, while it robustly models objects moving back to their original places, it is unable to predict when objects move to new locations for brief periods of time.
>
> [Guo 2019]  Guo, Xiaojie, et al. "Deep multi-attributed graph translation with node-edge co-evolution." 2019 IEEE International Conference on Data Mining (ICDM). IEEE, 2019.

---

### Meta-Review · Area_Chair_C21V · 2022-08-06

**Recommendation:** Accept (Poster)
**Confidence:** 4

**Metareview:**

The reviewers commended the paper on being well-written and proposing a new dataset on spatio-temporal dynamics of objects in households which could prove beneficial to the research community. The perspective on proactive robot assistance and object-centric modeling is interesting and relevant to the field. I encourage the authors to further address the remaining points raised in the reviews for the final version of the paper.

**Best Paper Nomination:**

No

---

> ### Author Response · Authors · 2022-08-25
> **Response to Meta Review by Area Chair C21V**
>
> We would like to thank the area chair and reviewers for all the feedback. We highlight some of our major responses here.
>
> **Performance and evaluation**
>
> - **Task performance**: Overall the task of predicting humans is difficult, and our evaluations include all household objects, many of which are not used very frequently. This emulates a more realistic setting but makes the prediction problem much harder. An example is when persona A (portrayed in figure 3e) misses breakfast on 21 of the 50 training days, and has breakfast late on 5 of the remaining 29 days. In such situations our system responds conservatively by not predicting cereal, bowl and milk to come out. As a result, our system relocates only 4% objects to wrong places, but leaves 58% of objects that could have been moved.
> - We have **added a confidence threshold parameter** that can further reduce the unnecessary movements and potentially raise the precision of 1-step predictions up to 89% and 2-step predictions up to 80%. We detail the results in our response to reviewer fzEn, along with **qualitative evaluations** of the system, both of which we will add to our revised paper.
> - **Evaluation in simulation**: To evaluate our system, we constructed a novel dataset based on crowdsourced human activity data to ensure that we could evaluate our approach in a longitudinal setting. Collecting many days of observations from multiple households in the real world would be very exciting, but is currently beyond our practical scope.
> - **Baselines**: We are not aware of any prior work on predictive modeling of object movements towards assistance, so we compare against other applications in robotics that require a similar underlying reasoning framework of object location tracking.
>
> **Justification of formulation**
>
> - We propose a novel formulation for proactive assistance, focusing on **long-horizon assistance** through object relocations at the **scale of an entire household**. Considering the difficulty of scaling action recognition to this domain, we propose a solution which **relies solely on object movement information**, and consider predictions several minutes ahead in time, keeping in mind practical considerations for household-scale mobile manipulation actions.
> - While we train personalized models to learn a household’s routine, we conducted additional experiments showing the benefit of **transferring pre-trained models** to a new household. The results are detailed in our response to reviewer QKuy, and will be added to our revised paper.